# Sperm Concentration Improvement May Be a Parameter Predicting Efficacy of FSH Therapy of Male Idiopathic Infertility

**DOI:** 10.3390/cells12182236

**Published:** 2023-09-08

**Authors:** Daniele Santi, Giorgia Spaggiari, Leonardo Dalla Valentina, Marilina Romeo, Federico Nuzzo, Lorenzo Serlenga, Laura Roli, Maria Cristina De Santis, Tommaso Trenti, Antonio R. M. Granata, Manuela Simoni

**Affiliations:** 1Department of Biomedical, Metabolic and Neural Sciences, University of Modena and Reggio Emilia, 41122 Modena, Italy; 2Unit of Endocrinology, Department of Medical Specialties, Azienda Ospedaliero-Universitaria of Modena, 41122 Modena, Italy; 3Unit of Andrology and Sexual Medicine of the Unit of Endocrinology, Department of Medical Specialties, Azienda Ospedaliero-Universitaria of Modena, 41122 Modena, Italy; 4Willis Towers Watson, Willis Holdings plc, London EC3M 7DQ, UK; 5Department of Laboratory Medicine and Pathology, Azienda USL of Modena, 41122 Modena, Italy

**Keywords:** male idiopathic infertility, FSH, sperm concentration, responders, total motile sperm number

## Abstract

Testis stimulation with follicle-stimulating hormone (FSH) is one of the empirical treatments proposed for male idiopathic infertility, although reliable markers to predict its efficacy are still lacking. This study aimed to identify parameters able to predict FSH efficacy in terms of pregnancy achievement. A real-world study was conducted, enrolling idiopathic infertile men treated with FSH 150IU three times weekly. Patients were treated until pregnancy achievement or for a maximum of two years and two visits were considered: V0 (baseline) and V1 (end of FSH treatment). Primary endpoints were the V1-V0 percentage change in sperm concentration, total sperm count, and total motile sperm number. In total, 48 pregnancies were recorded (27.7%) among 173 men (age 37.9 ± 6.2 years). All three endpoints increased after FSH administration, and only the V1-V0 percentage of sperm concentration significantly predicted pregnancy (*p* = 0.007). A V1-V0 sperm concentration of 30.8% predicted pregnancy, and the sperm concentration V1-V0 percentage (Y) required to obtain a pregnancy was predicted according to its baseline values (x): Y = 9.8433x^2^ − 203.67x + 958.29. A higher number of pregnancies was reached in men with baseline sperm concentration below 7.3 million/mL. Thus, the percentage of sperm concentration increasing after FSH administration could predict the treatment efficacy in terms of pregnancy. At the dosage used, the efficacy was significantly higher in patients with a starting sperm concentration < 7.3 mill/mL. Mathematical analyses identified a function able to predict the sperm concentration increase required to obtain a pregnancy in relation to the baseline sperm number.

## 1. Introduction

Male idiopathic infertility represents a wide category of infertile men in which the cause of spermatogenesis failure is still not detectable [1]. For this reason, available treatments are generally empirical and not evidence-based, including hormone administration [2], which is proposed to mimic the current treatment of male hypogonadotropic hypogonadism [3]. Indeed, in hypogonadotropic hypogonadism, the physiological hypothalamo–pituitary–gonadal axis is not functional, and much evidence demonstrated that the restoration of the stimulation of the testis through exogenous hormones is able to promote spermatogenesis, finally increasing pregnancy chances [3,4]. Several hormones could be used for this purpose, from the more physiological but little-used gonadotropin-releasing hormone (GnRH), to the easier-to-handle stimulation with follicle-stimulating hormone (FSH) and human chorionic gonadotropin (hCG) [3]. The reasoning behind this treatment is essentially that of a replacement therapy: where physiological hormones are lacking, they are therapeutically replaced to restore the natural regulation of spermatogenesis. A similar approach is empirically proposed in male idiopathic infertility, although in this clinical scenario, insufficient gonadotropins stimulation on the testis is not demonstrated [2,5]. In other words, a replacement treatment is used even in those patients in which the hormones are not quantitatively missing. Consequently, the actual efficacy of this empirical approach is limited, as suggested by 21 clinical trials available in the literature [6,7,8,9,10,11,12,13,14,15,16,17,18,19,20,21,22,23,24,25]. Collectively, these studies show that FSH administration to idiopathic infertile men increases the overall pregnancy rate, but the calculated number required to treat is elevated, suggesting that from 10 to 18 men should be treated to achieve one additional pregnancy [26]. Undeniably, this result indicates that FSH might not be the best treatment in case of idiopathic infertility, at least if administered with a replacement approach. The results may be different, shifting to an over-stimulatory approach [27]. A recent meta-analysis suggested that FSH administration increases sperm concentration and total sperm count in a dose-dependent manner [28]. FSH administration with an over-stimulating intent is already applied in clinical practice to women to obtain the largest possible number of oocytes to be used in assisted reproduction [29,30,31]. Thus, if a FSH-mediated supraphysiological stimulation is demonstrated efficient to increase gamete production in the female gonad, there are no endocrinological reasons why the same approach should not work in the male. 

To switch from replacement to an over-stimulating approach in the male setting, the most urgent need is to clarify how to measure FSH action, thus finding a pharmacodynamic marker of FSH efficacy. Then, the second step should be to identify a priori which subjects could be considered responders or non-responders to this treatment. These two issues remain without answer but are crucial in the therapeutic management of FSH for male idiopathic infertility. In a recent real-world data analysis evaluating idiopathic infertile men treated with FSH, we demonstrated that male partners of couples who achieved a pregnancy showed a sperm concentration and progressive sperm motility increase higher than equally treated men who did not reach pregnancy [32]. However, which sperm parameter improvement is necessary to reach a pregnancy? Are there any thresholds of sperm parameters’ change after FSH administration able to predict pregnancy? This retrospective clinical trial based on real-world data was designed to determine which semen parameters before and after FSH stimulation are able to distinguish male partners of pregnant couples from the others. As a secondary objective, we tried to identify, if possible, thresholds or mathematical functions able to measure the semen parameters improvement sufficient to obtain a pregnancy. 

## 2. Materials and Methods

### 2.1. Study Design

A single-center, retrospective, observational study was carried out based on real-world data. The original study was published elsewhere [32]. The current study started with a similar approach, increasing the sample size, considering all patients consecutively evaluated in the outpatient andrological clinic for couple infertility in Modena from June 2015 to May 2023. 

Inclusion criteria were: (i) male subjects, (ii) partner of infertile couple, (iii) meeting the eligibility criteria for the reimbursement of FSH required by note number 74 of the Agenzia Italiana del Farmaco (AIFA) [33], and (iv) treated with FSH. In detail, the AIFA note number 74 allows FSH reimbursement for those men with altered semen analysis and FSH basal levels below 8 IU/L. Only patients in which FSH administration was concluded were considered, i.e., (i) when pregnancy was achieved or (ii) after a maximum of one year of treatment without success. Thus, patients lost during the follow-up or still under FSH treatment were excluded.

Data collected during routinely performed visits were used to create a single dataset, in which two visits for each patient were included: the visit at baseline (V0) performed before FSH administration and the follow-up visit (V1) performed at the end of FSH treatment.

### 2.2. Andrological Diagnostic Work-Up

Data collected in the final dataset followed the standard diagnostic work-up provided for male infertility [34,35], as already described elsewhere [32]. 

When the patient fulfilled AIFA note 74, FSH was prescribed at the dosage of 150 IU three times weekly. Follow-up visits were scheduled after four months for a total of one year. During each visit, hormonal evaluations and semen analysis were performed. 

### 2.3. Data Collection

Among the data collected, the most relevant variables were represented by semen parameters obtained through conventional semen analyses, which were performed in the centralized laboratory of the Department of Clinical Pathology (Ospedale Civile of Baggiovara, Modena, Italy), following the World Health Organization (WHO) manual. The following semen parameters were considered: semen volume, sperm concentration, total sperm number, percentage of progressive and non-progressive motility, and percentage of sperm with normal morphology. On these data, the total motile sperm (TMS) number was calculated, multiplicating the percentage of progressive motile sperm by the total sperm number. 

The main endpoints of this study were the calculations of the percentage change from V0 to V1 (Δ) of the three most relevant parameters, i.e., sperm concentration, total sperm number, and TMS number. 

The number of clinical pregnancies obtained was recorded and used to divide patients a posteriori in the study (male partners of pregnant couples) and control group (male partners of non-pregnant couples).

### 2.4. Ethical

The study protocol was approved by the local ethics committee of the “Area Vasta Emilia Nord Modena” (protocol number AOU0024637/19 of 09/2019). Due to the retrospective design of the study, informed consent was not necessary.

### 2.5. Statistical Analysis

Each of the three main endpoints was evaluated separately. First, a descriptive analysis of Δ of semen analysis parameters after FSH treatment was performed. Second, the difference between the study and control groups was evaluated for each of the three main endpoints, using the Mann–Whitney *U*-test. Third, receiving operator curve (ROC) analysis was performed, using pregnancy as a state variable and the Δ of the three most relevant semen analysis parameters as test variables. Fourth, a logistic regression analysis was performed using pregnancy as a dependent and each of the three main endpoints as an independent variable separately. Finally, the distribution of the three main endpoints was evaluated against the baseline value of the semen parameter by visual inspection of the scatter plot. The most accurate function describing data dispersion was calculated, and the R-squared was used to evaluate the accuracy (Figure 1).

The “Statistical Package for the Social Science” software for Windows (version 28.0; SPSS Inc., Chicago, IL, USA) was used for statistical analyses. Statistical significance was considered for *p*-values < 0.05.

## 3. Results

One hundred and seventy-three subjects were included, and forty-eight pregnancies were recorded (27.7%) (Table 1).

### 3.1. Sperm Concentration

Considering the entire cohort, the overall Δ sperm concentration was 347.4 ± 1089.1%, without significant differences between study and control groups (433.6 ± 1076.1% versus 314.4 ± 1096.4%, *p* = 0.521). This result suggests that the evaluation of the simple Δ sperm concentration is not sufficient to predict pregnancy.

A ROC curve was generated considering pregnancy as a state variable and Δ sperm concentration as a test variable. The area under the curve (AUC) was 0.618 with *p* = 0.017 (confidence interval [CI]95%: 0.521–0.714). Thus, a Δ sperm concentration of 30.8% predicted pregnancy with 60.4% sensitivity and 60% specificity (Figure 2). 

This analysis highlighted that a Δ sperm concentration cut-off able to predict pregnancy could be suggested, although its accuracy remains low for potential clinical application.

Logistic regression analysis highlighted that Δ sperm concentration significantly predicted pregnancy (Wald 7.196, OR 1.001, 95%CI: 1.000, 1.002, *p* = 0.007). This result remained after adjustment for treatment duration (Wald 7.392, OR 1.001, 95%CI: 1.000, 1.002, *p* = 0.007), confirming the predictive role of this variable.

The Δ sperm concentration distribution was evaluated against baseline sperm concentration separately for the study and control group. Second-level polynomial functions best described the trend in both groups. Indeed, Δ sperm concentration in the study group was described by the following equation (R-squared 0.119) (Figure 3):Y = 9.8433x^2^ − 203.67x + 958.29,(1)
where x represents the baseline sperm concentration.

The Δ sperm concentration in the control group was described by the following equation (R-squared 0.196) (Figure 3):Y = 2.6624x^2^ − 58.725x + 282.69.

Figure 2 shows the Δ sperm concentration in both groups, with the green line representing the Δ sperm concentration distribution in the study group and the blue line in the control group (Figure 3). The vertical red line indicates the baseline sperm concentration threshold above which the two distributions overlap (i.e., 7.3 million/mL) (Figure 3). 

Since the second-order polynomial function significantly described the Δ sperm concentration (F 4.78, *p* = 0.013), the 5th and the 95th centiles of this regression analysis were used to calculate the two functions describing the confidence interval of this trend.
5th centile: Y = 1.7907x^2^ − 377.55x + 491.88,
95th centile: Y = 21.47x^2^ − 29.78x + 1424.69. 

These analyses indicate that treatment success in terms of pregnancy is expected when baseline sperm concentration is lower than 7.3 million/mL and when the Δ sperm concentration is included within the interval obtained by Y = 1.7907x^2^ − 377.55x + 491.88 and Y = 21.47x^2^ − 29.78x + 1424.69, where x represents the baseline sperm concentration.

By visual inspection of Figure 2, 67 patients (38.7%) did not increase sperm concentration after FSH administration. Thus, we divided the entire cohort of patients into two different subgroups, considering men who experienced a sperm concentration increase after FSH administration and those who did not. In the study group, the number of pregnancies was significantly higher in men who improved sperm concentration after FSH administration compared to the others (33.0% versus 19.0%, *p* = 0.037) (Table 2), suggesting that the pregnancy chance is related to sperm parameters improvement. However, all variables available were not significantly different between the two subgroups created considering control groups (Table 2). 

The patient cohort was then divided according to the sperm concentration threshold detected to determine when study and control group Δ sperm concentration distributions overlapped (i.e., 7.3 million/mL). In this way, we highlighted that a higher proportion of pregnancies was achieved in men with baseline sperm concentration below 7.3 million/mL compared to the other subjects (32.5% versus 17.8%, *p* = 0.032) (Table 3). Moreover, the two subgroups were significantly different in terms of Δ sperm concentration (*p* = 0.021 and *p* < 0.001, respectively), baseline progressive motility (*p* = 0.012), and FSH therapy duration (*p* = 0.004) (Table 3). 

Similarly, the Δ sperm concentration was significantly higher in the study compared to the control group only when men with baseline sperm concentrations below 7.3 million/mL were considered. Finally, in the subgroup of patients with baseline sperm concentration < 7.3 million/mL, the ROC analysis was repeated. The AUC was 0.61 with *p* = 0.047 (CI95%: 0.503–0.725), confirming that a threshold of Δ sperm concentration of 33.9% predicted pregnancy with 71.1% sensitivity and 58.2% specificity.

### 3.2. Total Sperm Count

A mean Δ total sperm number of 195.9 ± 577.5% was recorded in the overall cohort, showing no significant differences between study and control groups (160.8 ± 616.2% versus 208.5 ± 616.2%, *p* = 0.636). This result confirms that the Δ total sperm number is highly variable, with a high standard deviation and a scarce capability to predict the response to FSH stimulation. 

The ROC analysis showed low accuracy (AUC 0.45, 95%CI:0.343, 0.548), impeding the identification of a Δ total sperm number threshold to predict FSH efficacy (*p* = 0.281).

The Δ total sperm count distribution was described by the second-order polynomial function, with the following function in the study group (R-squared 0.0523) (Figure 4):Y = −0.0822x^2^ − 5.7225x + 261.3,
where x represents the baseline sperm concentration.

The Δ total sperm number distribution according to baseline levels in the control group was described by the following equation (R-squared 0.0032) (Figure 4):Y = −0.0427x^2^ + 1.7418x + 203.72.(2)

These two polynomial functions showed limited statistical relevance, as reported by the low R-squared and a substantial overlap (Figure 4), suggesting that total sperm count could not be considered a good parameter to predict FSH administration efficacy. Accordingly, logistic regression analysis was not able to demonstrate any predictive value of total sperm count (Wald 0.198, OR 1.000, 95%CI: 0.999, 1.001, *p* = 0.656).

### 3.3. Total Motile Sperm Number

A mean Δ TMS of 41.1 ± 250.7% was recorded after FSH administration, without a statistically significant difference between the study and control groups (207.0 ± 700.8% versus 41.1 ± 250.7%, *p* = 0.153). Similarly, the ROC analysis was not able to detect a Δ TMS threshold predicting pregnancy (AUC 0.42, 95%CI: 0.319, 0.531, *p* = 0.173).

The trend of Δ TMS was described by the following second-order polynomial functions in the study group (R-squared 0.0637) (Figure 5):Y = 1.1034x^2^ − 28.5x + 105.65,
where x represents the baseline sperm concentration.

Additionally, in the control group (R-squared 0.0265) (Figure 5):Y = 5.5136x^2^ − 90.3x + 295.66.

The functions describing the Δ TMS showed low accuracy and substantial overlap at visual inspection (Figure 5). Accordingly, logistic regression analysis failed to demonstrate a Δ TMS capability to predict pregnancy (Wald 0.122, OR 1.000, 95%CI: 0.999, 1.001, *p* = 0.727). This section may be divided by subheadings. It should provide a concise and precise description of the experimental results, their interpretation, as well as the experimental conclusions that can be drawn.

## 4. Discussion

Here, we demonstrate for the first time that FSH administration efficacy in terms of pregnancy achievement could be predicted by conventional semen parameters in male idiopathic infertility. In particular, not the sperm concentration per se, but its increase over baseline after FSH administration is able to predict pregnancy. In the clinical management of male idiopathic infertility, it appears evident that most FSH-treated men experience a variable degree of sperm concentration increase. However, this improvement does not always result in a pregnancy, as detected in one case out of four in our series. With this approach, we identified a mathematical function able to discriminate FSH-responders from non-responders through the Δ calculation of sperm concentration, intended as the difference between baseline and treatment values. Applying a conventional statistical analysis, we detected that an increase of 30.8% in sperm concentration resulted in predicting pregnancy, although with a suboptimal accuracy. Then, we evaluated the Δ sperm concentration distribution in study and control groups, showing that the two trends are divergent for men with baseline sperm concentrations below 7.3 million/mL. Thus, we calculated the confidence interval of the Δ sperm concentration distribution in the study group, identifying FSH-responders. With this in mind, we could speculate that an FSH-responder is a man with a baseline sperm concentration below 7.3 million/mL and with a Δ sperm concentration included between Y = 1.7907x^2^ − 377.55x + 491.88 and Y = 21.47x^2^ − 29.78x + 1424.69 (where x represents the baseline sperm concentration). This result was obtained by looking at the clinical practice and considering the current FSH application in male idiopathic infertility, but obviously, it requires further confirmation in specific research settings and/or in independent cohorts and might be different when using higher FSH dosages. We address the topic of FSH efficacy predictors with a more complex statistical analysis. Indeed, several attempts to determine potential predictors have been performed so far without conclusive remarks. Thus, we decided to face the challenge with a more complex approach, aiming to address the trend of FSH efficacy, although the result is still difficult to apply directly to clinical practice.

Although our study has a retrospective design, subgroup analyses are particularly interesting. Indeed, these further examinations confirmed that pregnancy chances are higher when baseline sperm concentration is lower than 7.3 million/mL. Moreover, pregnancies were achieved in this subgroup of men with a shorter duration of FSH administration, suggesting that a relatively fast sperm improvement could be expected. With this in mind, we could speculate that men with idiopathic infertility and sperm concentration below 7.3 million/mL are the best responders to the current FSH schedule, i.e., 150 IU three times weekly. In other words, the subgroup of subjects with the worst sperm concentration at baseline seems to be similar to hypogonadotropic hypogonadal men, who are known to respond well to FSH administration at a replacement dosage. As a confirmation, patients with baseline sperm concentration below 7.3 million/mL showed reduced sperm progressive motility at baseline compared to other men, suggesting that spermatogenesis in these patients is much more compromised compared to men with baseline sperm concentration higher than this threshold. This result confirms the potential analogy between these two different clinical conditions, suggesting that men with normal FSH levels and sperm concentration < 7.3 million/mL can be regarded as patients with “functional” hypogonadotropic hypogonadism. On the contrary, we could speculate that men with baseline sperm concentrations higher than 7.3 million/mL might respond to an over-stimulatory approach. Indeed, in this second subgroup, the Δ sperm concentration after treatment was similar compared to the study and control groups. These patients should be enrolled in future studies in which higher FSH dosages should be applied. Overall, our work describes two different populations within the same cohort of men with idiopathic infertility, confirming the known high heterogeneity of this clinical condition. Future studies in this field must consider this heterogeneity when assessing FSH efficacy.

This is not the first attempt to correlate semen parameters with pregnancy. In the literature, several studies attempted to identify a potential predictive role of seminal data in couple infertility, although without definitive results. Old studies suggested semen parameters as potential predictors of time to pregnancy (TTP) [36], although this association failed to be confirmed subsequently [37,38,39,40]. Larger cohorts have been recently evaluated, showing promising results. A recent longitudinal cohort study of over 6000 couples found that higher sperm concentration, total sperm count, progressive motility and TMS were associated with better chances of conception during a five-year period and a shorter time to conception [41]. A TMS threshold of 50 million in the overall cohort and of 20 million in the natural conception cohort were identified to best predict a pregnancy within five years [29]. Accordingly, over 90,000 intra-uterine insemination (IUI) cycles were evaluated in a coeval study, showing that pregnancy rates were the highest when TMS was higher than 9 million [42]. A large real-world data analysis on 22,013 assisted reproduction cycles showed that sperm motility plays a role in predicting in vitro fertilization success, while sperm morphology resulted in the relevant parameter in intracytoplasmic sperm injection cycles [43]. All these results were not confirmed by a recent meta-analysis in which 17 studies were combined, showing no significant associations between sperm concentration, motility, or morphology and clinical pregnancy rate [44]. Thus, the bulk of current evidence suggests that individual semen parameters could have a role in the prediction of pregnancy, although their statistical power is still limited and requires a large number of samples. Among influencing factors, it should be considered that all these results are heterogeneous, evaluating mixed clinical conditions and generally not considering all factors that could be related to pregnancy, such as sperm quality (i.e., sperm DNA fragmentation index), potential treatments for the male partner, and all variables related to female fertility/infertility treatment. Comprehensively, a clear identification of which should be the most relevant male parameter able to measure/monitor male fertility status is still lacking. In this complex scenario, even less evidence is available on the best efficacy marker of FSH treatment in male idiopathic infertility. Several studies reported an average increase in sperm concentration in response to FSH administration [45], while in other studies, a reduction in sperm DNA fragmentation index emerged [13,46]. However, this latter parameter, which has recently been promoted as an additional in the latest WHO manual [47], is still far from being used in clinical practice, and data are limited to research settings. Although the identification of the most appropriate marker of FSH efficacy is outside the scope of our study, it is undeniable that this issue remains the watershed in the FSH therapeutic management in male idiopathic infertility. In addition, no clinical parameters are currently applied in clinical practice to distinguish a priori FSH-responders from non-responders; despite some attempts involving testosterone serum levels [48], inhibin B serum levels [49], spermatid count [50], and pharmacogenetics markers (i.e., FSHβ or FSH receptor polymorphisms) [20,24], this remains unconfirmed. 

## 5. Conclusions

Our study has several limitations, notably the retrospective design and the limited sample size. On the other hand, we enrolled a relatively homogeneous population, evaluated in a single center, who performed all hormonal and semen examinations in the same laboratory. Moreover, our study had the main virtue of describing the trend of seminal parameters after FSH-administration in an objective way for the first time, suggesting that a prediction of pregnancy is possible starting from routinely performed conventional semen analyses.

## Figures and Tables

**Figure 1 cells-12-02236-f001:**
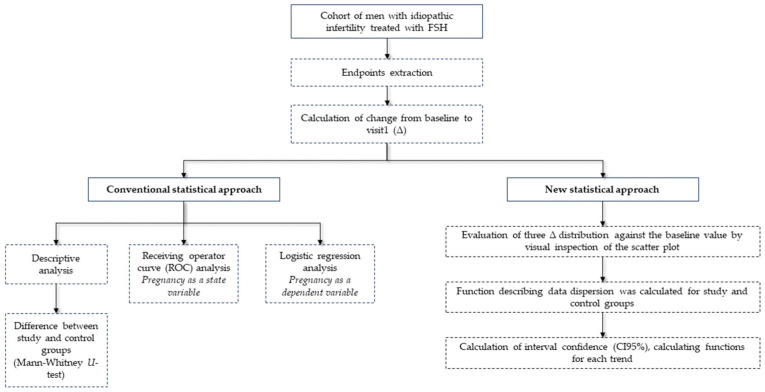
Flow chart to explain the statistical analysis approach.

**Figure 2 cells-12-02236-f002:**
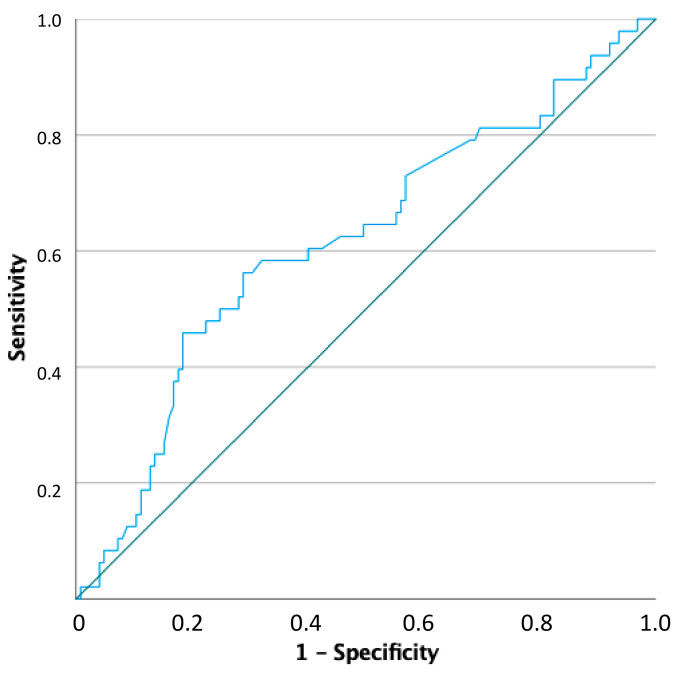
Receiving operator curve (ROC) analysis using pregnancy as test variable and the percentage of sperm concentration increase as state variable. The green line is the intercept, the blue line the one calculated.

**Figure 3 cells-12-02236-f003:**
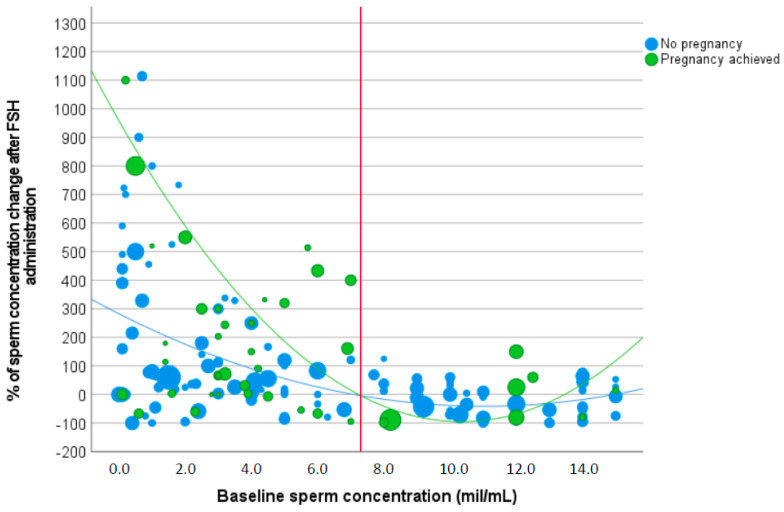
Scatter plot of percentage of sperm concentration change after follicle-stimulating hormone (FSH) administration versus baseline sperm concentration. The dimension of each point is related to the duration of FSH stimulation. The vertical red line indicates the baseline sperm concentration threshold above which the two distributions overlap (i.e., 7.3 million/mL) (Footnotes to Figure 2: FSH = follicle-stimulating hormone).

**Figure 4 cells-12-02236-f004:**
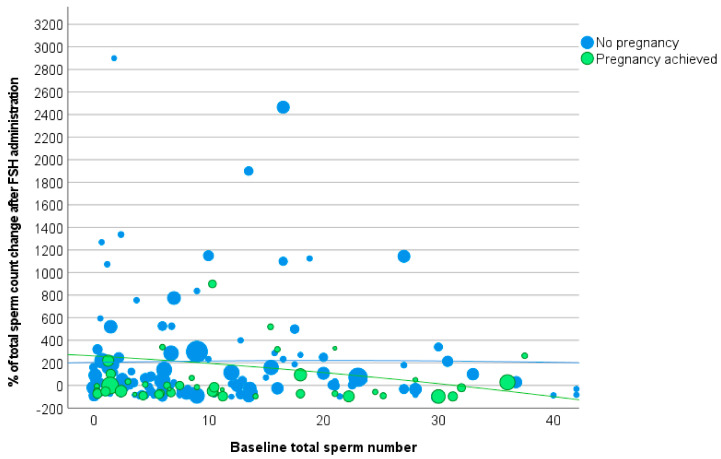
Scatter plot of percentage of total sperm count change after follicle-stimulating hormone (FSH) administration versus total sperm count at baseline. The dimension of each point is related to the duration of FSH stimulation (Footnotes to Figure 3: FSH = follicle-stimulating hormone).

**Figure 5 cells-12-02236-f005:**
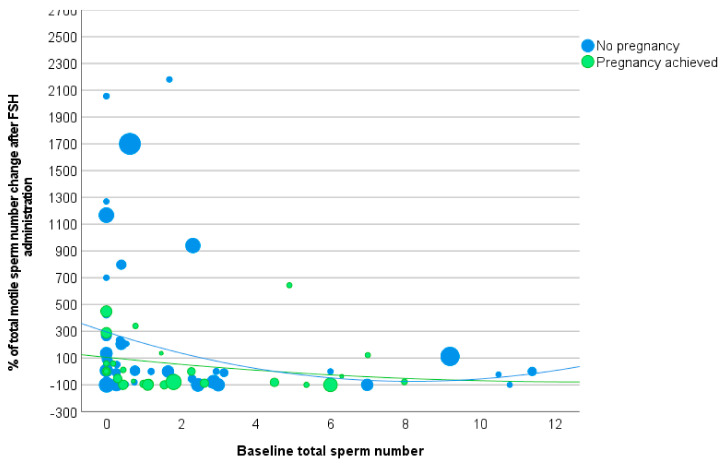
Scatter plot of percentage of total motile sperm number change after FSH administration versus total motile sperm number at baseline. The dimension of each point is related to the duration of FSH stimulation.

**Table 1 cells-12-02236-t001:** Baseline characteristics of follicle-stimulating hormone (FSH)-treated patients. Continuous data are reported as mean ± standard deviation (minimum, maximum).

Variables	FSH-Treated Patients (n = 173)
Age (years)	37.9 ± 6.2 (23.3–51.4)
Infertility duration (years)	2.4 ± 3.1
Primary infertility n (%)	156 (90.2)
Total testosterone (ng/mL)	5.5 ± 2.7 (2.1–11.8)
Luteinising hormone (LH) (IU/L)	3.5 ± 1.4 (1.7–9.5)
Follicle-stimulating hormone (FSH) (IU/L)	4.4 ± 2.0 (1.0–7.9)
Prolactin (ng/mL)	12.9 ± 6.1 (1.5–30.9)
Semen volume (mL)	2.5 ± 1.6 (0.3–11.5)
Sperm concentration (million/mL)	5.5 ± 4.7 (0.1–14.9)
Progressive sperm motility (%)	18.6 ± 16.3 (0.0–71.0)
Sperm with normal morphology (%)	2.7 ± 5.6 (0.0–17.0)
Duration of FSH administration (months)	8.0 ± 4.5 (2.0–12.0)
Pregnancies obtained n (%)	48 (27.7)

**Table 2 cells-12-02236-t002:** Characteristics of follicle-stimulating hormone (FSH)-treated patients dividing study and control group and comparing patients who increased (n = 71) to those who did not increase (n = 54) sperm concentration after treatment. Continuous data are reported as mean ± standard deviation and compared with the Mann–Whitney *U*-test. Categorical data are expressed as numbers (%) and compared with the Fisher Exact test.

Variables	Study Group (n = 48)	Control Group (n = 125)	*p*-Value
Age (years)			
*Increased sperm concentration*	35.7 ± 4.9	39.0 ± 6.9	0.064
*Not increased sperm concentration*	37.4 ± 4.7	37.9 ± 6.0	0.774
*p*-value	0.305	0.343	
Total testosterone (ng/mL)			
*Increased sperm concentration*	5.7 ± 3.0	5.5 ± 2.4	0.750
*Not increased sperm concentration*	5.0 ± 1.6	5.5 ± 3.2	0.600
*p*-value	0.509	0.882	
Luteinising hormone (LH) (IU/L)			
*Increased sperm concentration*	3.8 ± 1.0	3.4 ± 1.3	0.292
*Not increased sperm concentration*	4.3 ± 2.2	3.3 ± 1.3	0.091
*p*-value	0.374	0.701	
Follicle-stimulating hormone (FSH) (IU/L)			
*Increased sperm concentration*	4.4 ± 1.4	4.3 ± 2.1	0.848
*Not increased sperm concentration*	5.0 ± 2.1	4.5 ± 2.0	0.491
*p*-value	0.332	0.589	
Prolactin (ng/mL)			
*Increased sperm concentration*	13.1 ± 4.1	12.6 ± 6.1	0.824
*Not increased sperm concentration*	9.3 ± 4.7	13.8 ± 7.0	0.181
*p*-value	0.120	0.538	
Baseline semen volume (mL)			
*Increased sperm concentration*	2.3 ± 1.3	2.5 ± 1.7	0.647
*Not increased sperm concentration*	3.5 ± 1.7	2.3 ± 1.8	0.065
*p*-value	**0.018**	0.704	
Δ Total sperm count			
*Increased sperm concentration*	204.0 ± 939.5	237.2 ± 507.1	0.809
*Not increased sperm concentration*	57.5 ± 300.9	169.4 ± 441.5	0.391
*p*-value	0.568	0.443	
Baseline progressive sperm motility (%)			
*Increased sperm concentration*	21.1 ± 17.8	17.4 ± 17.0	0.391
*Not increased sperm concentration*	21.6 ± 10.6	17.6 ± 16.0	0.429
*p*-value	0.933	0.957	
Δ total motile sperm number (%)			
*Increased sperm concentration*	236.0 ± 939.5	237.2 ± 594.9	0.992
*Not increased sperm concentration*	13.8 ± 238.5	26.3 ± 222.9	0.882
*p*-value	0.490	0.086	
Baseline sperm with normal morphology (%)			
*Increased sperm concentration*	4.9 ± 11.8	2.0 ± 2.7	0.237
*Not increased sperm concentration*	2.6 ± 4.0	2.3 ± 2.7	0.761
*p*-value	0.610	0.749	
Duration of FSH administration (months)			
*Increased sperm concentration*	7.4 ± 4.2	8.1 ± 4.6	0.487
*Not increased sperm concentration*	7.9 ± 5.4	8.2 ± 4.3	0.814
*p*-value	0.751	0.854	
Patients’ distribution n (%)			
*Increased sperm concentration*	35 (20.2)	71 (41.0)	-
*Not increased sperm concentration*	13 (7.5)	54 (31.2)	-
*p*-value	-	-	**0.037**

**Table 3 cells-12-02236-t003:** Characteristics of follicle-stimulating hormone (FSH)-treated patients dividing study and control group and comparing patients with baseline sperm concentration below (n = 117) or above (n = 56) 7.3 million/mL Continuous data are reported as mean ± standard deviation and compared with the Mann–Whitney *U*-test. Categorical data are expressed as numbers (%) and compared with the Fisher Exact test.

Variables	Study Group (n = 48)	Control Group (n = 125)	*p*-Value
Age (years)			
*Baseline sperm concentration < 7.3 million/mL*	36.5 ± 5.0	37.6 ± 6.2	0.347
*Baseline sperm concentration > 7.3 million/mL*	35.0 ± 4.2	40.2 ± 6.9	0.088
*p*-value	0.409	0.082	
Total testosterone (ng/mL)			
*Baseline sperm concentration < 7.3 million/mL*	5.4 ± 2.6	5.3 ± 2.8	0.832
*Baseline sperm concentration > 7.3 million/mL*	5.5 ± 2.7	5.9 ± 2.8	0.728
*p*-value	0.948	0.324	
Luteinising hormone (LH) (IU/L)			
*Baseline sperm concentration < 7.3 million/mL*	3.9 ± 1.6	3.5 ± 1.4	0.283
*Baseline sperm concentration > 7.3 million/mL*	4.1 ± 1.0	3.3 ± 1.2	0.065
*p*-value	0.648	0.524	
Follicle-stimulating hormone (FSH) (IU/L)			
*Baseline sperm concentration < 7.3 million/mL*	4.4 ± 1.6	4.3 ± 2.0	0.828
*Baseline sperm concentration > 7.3 million/mL*	5.0 ± 1.7	4.5 ± 2.1	0.506
*p*-value	0.377	0.699	
Prolactin (ng/mL)			
*Baseline sperm concentration < 7.3 million/mL*	12.5 ± 3.4	12.8 ± 6.6	0.868
*Baseline sperm concentration > 7.3 million/mL*	11.2 ± 5.9	13.7 ± 6.5	0.362
*p*-value	0.584	0.639	
Baseline semen volume (mL)			
*Baseline sperm concentration < 7.3 million/mL*	2.6 ± 1.5	2.6 ± 1.8	0.983
*Baseline sperm concentration > 7.3 million/mL*	2.9 ± 1.3	2.1 ± 1.5	0.185
*p*-value	0.619	0.137	
Δ sperm concentration (%)			
*Baseline sperm concentration < 7.3 million/mL*	445.0 ± 781.8	164.5 ± 315.1	**0.007**
*Baseline sperm concentration > 7.3 million/mL*	−9.8 ± 79.1	−16.8 ± 59.3	0.752
*p*-value	**0.021**	**<0.001**	
Δ Total sperm count			
*Baseline sperm concentration < 7.3 million/mL*	174.8 ± 853.2	203.2 ± 465.8	0.818
*Baseline sperm concentration > 7.3 million/mL*	124.3 ± 411.1	217.0 ± 507.2	0.592
*p*-value	0.857	0.879	
Baseline progressive sperm motility (%)			
*Baseline sperm concentration < 7.3 million/mL*	17.6 ± 14.7	15.0 ± 15.0	0.465
*Baseline sperm concentration > 7.3 million/mL*	32.3 ± 13.5	21.0 ± 17.9	0.083
*p*-value	**0.012**	0.087	
Δ total motile sperm number (%)			
*Baseline sperm concentration < 7.3 million/mL*	198.5 ± 921.7	200.5 ± 578.9	0.990
*Baseline sperm concentration > 7.3 million/mL*	136.1 ± 362.0	60.8 ± 247.6	0.479
*p*-value	0.853	0.175	
Baseline sperm with normal morphology (%)			
*Baseline sperm concentration < 7.3 million/mL*	1.6 ± 1.9	1.8 ± 2.2	0.832
*Baseline sperm concentration > 7.3 million/mL*	4.7 ± 5.9	2.5 ± 3.0	0.098
*p*-value	0.065	0.333	
Duration of FSH administration (months)			
*Baseline sperm concentration < 7.3 million/mL*	6.7 ± 3.7	8.3 ± 4.5	0.065
*Baseline sperm concentration > 7.3 million/mL*	11.7 ± 6.2	8.0 ± 4.4	0.111
*p*-value	**0.004**	0.720	
Patients’ distribution n (%)			
*Baseline sperm concentration < 7.3 million/mL*	38 (22.0)	79 (45.7)	-
*Baseline sperm concentration > 7.3 million/mL*	10 (5.8)	46 (26.6)	-
*p*-value	-	-	**0.032**

## Data Availability

Data are available after contact with the corresponding author.

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
