# Peer review of "Sperm Concentration Improvement May Be a Parameter Predicting Efficacy of FSH Therapy of Male Idiopathic Infertility"

_cells, 2023, doi:10.3390/cells12182236_

Round 1

Reviewer 1 Report

The authors demonstrate that FSH administration efficacy of pregnancy achievement could be predicted by conventional semen parameters in male idiopathic infertility. In particular, not the sperm concentration per se, but its increase over baseline after FSH administration is able to predict pregnancy.  I am not certain that the data presentation involving extensive, and complex statistical analysis will convince the average practitioner that the FSH treatment is relevant treatment. Otherwise, while the report is interesting, it doesn't add significant clinical data to the field.

the authors need to explain for the casual reader why the complex statistical analysis is necessary to support the conclusions reached.  Does this make the treatment a useful approach to overcome infertility?  

no comments

Author Response

Point 1: The authors demonstrate that FSH administration efficacy of pregnancy achievement could be predicted by conventional semen parameters in male idiopathic infertility. In particular, not the sperm concentration per se, but its increase over baseline after FSH administration is able to predict pregnancy.  I am not certain that the data presentation involving extensive, and complex statistical analysis will convince the average practitioner that the FSH treatment is relevant treatment. Otherwise, while the report is interesting, it doesn't add significant clinical data to the field.

The authors need to explain for the casual reader why the complex statistical analysis is necessary to support the conclusions reached.  Does this make the treatment a useful approach to overcome infertility? 

Response 1: Thank you for your comment. As we reported in the manuscript, many attempts to evaluate the FSH efficacy in male idiopathic infertility has been performed so far. Some trials detected an efficacy in terms of pregnancy but were not able to identify potential predictors of this success. Is this failure to detect predictors caused by the sample size analysed so far or by the conventional statistical analyses generally applied? Since we are not able to answer this question, we decided to apply more complex statistical analysis to find out this potential effect. In particular, the idea that the FSH efficacy could be different considering the baseline sperm quality has been suggested by our daily routine clinical practice. Thus, we decided to improve the analyses of data, trying to highlight this dynamic trend. We added this issue within the ‘Discussion’ section, lines 319-322.

Reviewer 2 Report

Authors make an interesting investigation about the use of mathematical functions able to predict pregnancy rate from semen parameters of infertile men after FSH administration.

Overall, the manuscript is well written, structured and presented. A few modifications should be revised:

A figure that describe and explain the study design should be added to material and methods section.

Line 151-152: authors describe some results with numerical data which are not presented in a table or in a figure in the current manuscript.

Manuscripts contain some grammaratical errors, minor English revision is required.

Author Response

Point 1: Authors make an interesting investigation about the use of mathematical functions able to predict pregnancy rate from semen parameters of infertile men after FSH administration. Overall, the manuscript is well written, structured and presented. A few modifications should be revised:

A figure that describe and explain the study design should be added to material and methods section.

Response 1: Thank you for your suggestion. We added a figure highlighting the statistical flow applied.

Point 2: Line 151-152: authors describe some results with numerical data which are not presented in a table or in a figure in the current manuscript.

Response 2: Thank you for your comment. We reported the Δ change of the three main semen parameters as data within the text and not tables, in order to not create a new table that could create confusion to the reader.